# Impact of Sexual Activity on the Risk of Male Genital Tumors: A Systematic Review of the Literature

**DOI:** 10.3390/ijerph18168500

**Published:** 2021-08-11

**Authors:** Felice Crocetto, Davide Arcaniolo, Luigi Napolitano, Biagio Barone, Roberto La Rocca, Marco Capece, Vincenzo Francesco Caputo, Ciro Imbimbo, Marco De Sio, Francesco Paolo Calace, Celeste Manfredi

**Affiliations:** 1Urology Unit, Department of Neurosciences, Reproductive Sciences and Odontostomatology, University of Naples “Federico II”, 80121 Naples, Italy; felice.crocetto@gmail.com (F.C.); biagio193@gmail.com (B.B.); robertolarocca87@gmail.com (R.L.R.); drmarcocapece@gmail.com (M.C.); vincitor@me.com (V.F.C.); ciro.imbimbo@unina.it (C.I.); frap.calace@gmail.com (F.P.C.); manfredi.celeste@gmail.com (C.M.); 2Urology Unit, Department of Woman Child and of General and Specialist Surgery, University of Campania “Luigi Vanvitelli”, 80121 Naples, Italy; davide.arcaniolo@gmail.com (D.A.); marco.desio@unicampania.it (M.D.S.)

**Keywords:** sexual activity, sex, prostate cancer, penile cancer, testicular cancer

## Abstract

Most cancers are related to lifestyle and environmental risk factors, including smoking, alcohol consumption, dietary habits, and environment (occupational exposures). A growing interest in the association between sexual activity (SA) and the development of different types of tumors in both men and women has been recorded in recent years. The aim of the present systematic review is to describe and critically discuss the current evidence regarding the association between SA and male genital cancers (prostatic, penile, and testicular), and to analyze the different theories and biological mechanisms reported in the literature. A comprehensive bibliographic search in the MEDLINE, Scopus, and Web of Science databases was performed in July 2021. Papers in the English language without chronological restrictions were selected. Retrospective and prospective primary clinical studies, in addition to previous systematic reviews and meta-analyses, were included. A total of 19 studies, including 953,704 patients were selected. Case reports, conference abstracts, and editorial comments were excluded. Men with more than 20 sexual partners in their lifetime, and those reporting more than 21 ejaculations per month, reported a decreased risk of overall and less aggressive prostate cancer (PCa). About 40% of penile cancers (PCs) were HPV-associated, with HPV 16 being the dominant genotype. Data regarding the risk of HPV in circumcised patients are conflicting, although circumcision appears to have a protective role against PC. Viral infections and epididymo-orchitis are among the main sex-related risk factors studied for testicular cancer (TC); however, data in the literature are limited. Testicular trauma can allow the identification of pre-existing TC. SA is closely associated with the development of PC through high-risk HPV transmission; in this context, phimosis appears to be a favoring factor. Sexual behaviors appear to play a significant role in PCa pathogenesis, probably through inflammatory mechanisms; however, protective sexual habits have also been described. A direct correlation between SA and TC has not yet been proven, although infections remain the most studied sex-related factor.

## 1. Introduction

Sexual activity (SA) has long been a popular topic in different fields of medicine due to its potential impact on health status. SA includes a variety of activities, such as penetrative sex (anal and vaginal), oral sex, and masturbation, which could affect the psychophysical condition of men [1]. The effect of SA on the health status and, in particular, on the risk of developing tumors, has not been well clarified. A portion of the literature defines SA as a panacea, resulting in a positive influence on more than one domain of health status. In this context, Cao et al. showed that SA may reduce the risk of fatal coronary events and breast cancer, improving the quality of life, well-being, and cognitive function [2]. In contrast, other research has highlighted the negative aspects of SA, focusing on the risk of contracting sexually transmitted diseases (STDs), which may be the primum movens of different types of cancer. For example, human papillomavirus (HPV) infection is known to be a key risk factor for cervical tumors and a recognized risk factor for cancer of the penis, vulva, vagina, anus, and oral cavity [3]. Another essential aspect to consider in this context is the possible inflammation that can result, directly or indirectly, from SA, which is a critical component of the tumor pathway. The available literature reports that chronic inflammation increases the risk for several cancers through a variety of mechanisms involving the tumor microenvironment [4]. More specifically, it is hypothesized that inflammatory injury may favor carcinogenesis by causing cellular stress and genomic damage via high production of ROS [5]. Evidence from a number of genetic, epidemiological, and molecular studies suggests that inflammation plays a crucial role in various stages of prostatic carcinogenesis and tumor progression [6]. The aim of the present systematic review is to describe and critically discuss the current evidence regarding the association between SA and male genital tumors (prostatic, penile, and testicular), focusing on the different theories and biological mechanisms reported in the literature.

## 2. Materials and Methods

This analysis was conducted and reported according to the general guidelines recommended by the Primary Reporting Items for Systematic Reviews and Metanalyses (PRISMA) statement [7].

The bibliographic search was performed in the MEDLINE (US National Library of Medicine, Bethesda, MD, USA), Scopus (Elsevier, Amsterdam, The Netherlands), and Web of Science (Thomson Reuters, Toronto, ON, Canada) databases in July 2021, without chronological restrictions. The following terms were combined in a title–abstract search to capture all relevant publications: (“sex” OR “sexual activity” OR “intercourse” OR “masturbation” OR “anal sex” OR “oral sex”) AND ((“prostate” OR “prostatic”) OR (“testis” OR “testicular”) OR (“penis” OR “penile”)) AND (“cancer” OR “tumor” OR “neoplasm”, OR “lesion”). All articles identified from the literature search were screened by two independent reviewers (B.B. and V.F.C.), with any discrepancies resolved by a third author (D.A.). The reference lists of included papers were used to search for other relevant articles.

To assess the eligibility of the articles, PICOS (participants, intervention, comparison, outcomes, study type) criteria were used [8]. PICOS criteria were set as follows: participants—male patients; intervention—any form of sexual activity; comparison—not applicable; outcome—cancer; study types—prospective and retrospective studies (controlled and uncontrolled), systematic reviews, and meta-analyses. Only articles in English describing at least 10 patients were selected. Preclinical studies exploring pathophysiology mechanisms of cancer development were included. Conference abstracts and commentaries were excluded.

The results were narratively reported, without performing a quantitative synthesis of the data, due to the expected high heterogeneity of the studies. The following data were extracted from clinical study: first author, publication year, study design, sample size, and main findings. The quality of the included studies was assessed using the Newcastle–Ottawa Scale (NOS) for non-randomized studies. Systematic reviews and meta-analyses were analogously assessed using A Measurement Tool to Assess Systematic Reviews-2 (AMSTAR-2). Finally, the risk of bias was assessed using the ROBINS-I Tool for each of non-randomized and randomized studies [9,10,11]. Ethical approval and patients’ consent were not required for the present study.

## 3. Results

The search strategy revealed a total of 390 results. Screening of the titles and abstracts revealed 306 papers that were eligible for inclusion. Further assessment of eligibility, based on full-text articles, led to the exclusion of 287 papers. Finally, 19 studies involving a total of 953,704 patients were included in the final analysis [12,13,14,15,16,17,18,19,20,21,22,23,24,25,26,27,28,29,30] (Figure 1). The main characteristics of the selected studies are summarized in Table 1, and the assessment of the risk of bias is reported in Figure 2 and Figure 3.

### 3.1. Prostate Cancer (PCa)

PCa is one of the most diagnosed cancers globally, with an estimated incidence of 1,276,000 new cancer cases and 359,000 deaths in 2018. Several risk factors have been established (e.g., old age, family history, and African American ethnicity) [31,32,33]. PCa is the neoplasia that has the greatest impact on the male population and is the most frequently detected cancer globally for men over 50 years of age. Despite recent therapeutic advances, PCa still represents a major urological disease associated with substantial morbidity and mortality. The most important role in the development and maintenance of normal male physiology is mediated by androgens. In particular, testosterone is a sex hormone that plays important roles in the body via the libido, bone mass, fat distribution, muscle strength, and mass, and also regulates spermatogenesis [34]. Recently, evidence has suggested that sexual behaviors, such as number of sexual partners, sexual orientation, ejaculation frequency (EF), gender of sexual partners, and impact of STDs, may play a role in PCa pathogenesis. Spence et al. showed that men with more than 20 sexual partners (females and males combined) in their lifetime had a decreased risk of overall and less aggressive PCa (OR 0.78, 95% CI 0.61–1.00 and OR 0.75, 95% CI 0.57–0.99, respectively) [12]. In addition, men considering themselves to be homosexual or bisexual had a slightly greater PCa risk, compared to men who self-identified as heterosexual. In particular, several male sexual partners were associated with an increased risk of PCa [12]. The pathophysiology is not clear but it is thought that physical trauma of the prostate during receptive anal intercourse and the major incidence of HIV could favor the development of PCa [35]. Rider et al. showed that men reporting ≥ 21 ejaculations per month compared to subjects with 4–7 ejaculations per month had a significantly lower risk of total PCa, with an HR of 0.81 (95% CI 0.72–0.92) and 0.78 (95% CI 0.69–0.89) at 20–29 years and 40–49 years, respectively [13]. Papa et al. described an inverse association between EF and PCa at age 30 to 39 (OR per 5-unit increase per week 0.83, 95% CI: 0.72–0.96) but not at ages 20 to 29 or 40 to 49 [14]. Several hypotheses were proposed to explain the correlation between EF and PCa. In the prostatic acini, luminal secretion is rich in corpora amylacea, and the increase in their number with age appears to be related to an increased incidence of PCa. A greater level of sexual activity may lead to less accumulation of these bodies, which has been supposed to have a carcinogenic effect [36,37]. Furthermore, EF regulates the expression of several genes in the prostate tissue. Sinnot et al. reported that EF modifies some molecular pathways, such as Ubiquitin Mediated Proteolysis (involved in cell cycle regulation), and the Citrate Cycle pathway (involved in citrate production). Citrate production decreases in PCa and increases in benign prostatic hyperplasia (BPH) [15]; moreover, it is known that in early prostate tumorigenesis there is a metabolic switch from citrate secretion to citrate oxidation [38]. Regarding sexual orientation, in 2002 Rosenblatt et al. showed that there was no association between sexual orientation and PCa [16]. Finally, EF may be linked to the alleviation of psychological tension and thus to the suppression of the central sympathetic nervous system, which may reduce the stimulation of the division of the epithelial cells of the prostate [17]. In addition, increased sexual activity can also be considered a risk factor for PCa development. Multiple and lasting episodes of STDs may represent a high risk for PCa development. It was proposed that gonorrhea and other bacterial infections may cause PCa through prostate inflammation and atrophy, whereas viral infections may have direct pro-oncogenetic properties or immunosuppressive effects [18]. In a meta-analysis performed by Lian et al., involving 21 studies and 9965 patients with PCa, the authors found a high incidence of tumors in patients with gonorrhea infection (OR 1.31, 95% CI 1.14–1.52), particularly in African American males (OR 1.32, 95% CI 1.06–1.65) [39]. Data showed that HIV is associated with several genitourinary malignancies with an increased incidence in the population with HPV infection [40]. Hessol et al. identified 32 cases of PCa in a population of 14,000 adults in San Francisco between 1990 and 2001. The standardized incidence was greater when compared to the general population [41]. A recent meta-analysis by Sun et al., evaluating 27 articles and 2780 men with HIV/AIDS, showed a decreased incidence of PCa in patients with HIV infection (SIR 0.76; 95% CI, 0.64–0.91; *p* = 0.003) [19]. Nevertheless, a significant correlation between PCa incidence and duration of HIV infection (*p* = 0.047) has been previously demonstrated [20]. Among viruses, human papillomavirus (HPV) infection is related to several cancers and is the most important oncogenic virus. About 15% of all human cancers are caused by HPV and, specifically, more than 60,000 cases of cervix, penile, vaginal, vulvar, anal, neck, and head cancers [42]. Chronic inflammation is the pathogenetic mechanism related to cancer. There are more than 100 types of HPV; however, HPV 16 and HPV 18 are the most common oncogenetic forms [43]. Although there are more than 200 types, only 40 are related to genital tract infection. These types can be divided into two categories: low-risk HPV types (6, 11, 42, 43, and 44), causing genital warts; and high-risk HPV types (16, 18, 31, 33, 39, etc.), causing cancer. HPV is the most common sexually transmitted virus, although non-sexual transmission is also reported. Early beginning of sexual activity, several sexual partners, use of oral contraceptives, low socioeconomic status, and smoking habit were found to be the most important risk factors that contribute to HPV infection [44]. HPV infects basal epithelial cells and, through the E6 and E7 oncoprotein, leads to cell transformation: E6 binds and inactivates the p53 pathway, whereas E7 binds and inactivates pRb, p107, and p130 tumor suppressor gene products. p53 and pRb are involved in cellular tumor suppressor processes, such as cell cycle progression, DNA repair, apoptosis, differentiation, senescence, and chromatin remodeling [45,46] (Figure 4).

These two oncoproteins are able to bind p53, Bcl-2/surviving and retinoblastoma proteins, respectively, which are related to the cellular cycle control. In a retrospective study conducted in 2017, Glenn et al. identified HPV in 28 of 52 patients with an initial benign prostate biopsy and who subsequently developed PCa. This virus was biologically active and the higher prevalence of E7 protein suggested that HPV oncogenic activity was an early phenomenon in prostate oncogenesis [47]. Yin et al., in a metanalysis of 24 studies involving 971 patients and 1085 controls, showed that HPV infection significantly increased the risk of PCa (OR = 2.27 (95% CI, 1.40–3.69) [48].

### 3.2. Penile Cancer (PC)

Although PC is an uncommon tumor, its incidence is significantly higher in developing countries [49]. Globally, 26,300 new cases of penile cancer are diagnosed each year [50]. Its prevalence varies between different populations, and the current opinion is that these race effects are related to differences in sexual and behavior patterns [51]. Squamous cell carcinoma (SCC) represents the majority of cases of PC, and up to 80% of cases are localized at the glans and prepuce [52]. Penile cancer and its treatment may have a significant impact on sexuality and intimacy, body image, urinary function, mental health, and health-related quality of life (HRQOL) [53]. The recognized risk factors for PC include poor penile hygiene, HPV infection, smoking, and chronic inflammatory conditions [54]. In particular, HPV infection can result in a spectrum of genitourinary manifestations, ranging from genital warts to PC [21]. Nationally representative surveys conducted in the USA showed that HPV may achieve a prevalence of around 80% depending on parameters of the population including age, sex, immune status, and sexual behavior [22]. To date, over 200 HPV genotypes have been identified, which can be classified according to their oncogenicity in high-risk genotypes, including HPV 16, 18, 33, and 35, whereas low-risk genotypes include HPV 6 and 11 [55]. In about 50% of cases, multiple genotypes are present at the same time. Although low-risk genotypes are the main cause of genital warts, they are generally not related to PC [56]. In a recent systematic review, Parkin and Bray found that about 40% of PCs were related to HPV, and the most frequent genotype was HPV 16. However, they also noted that this percentage may be overestimated by a coincident presence of the virus, rather than a causal relationship [50]. The HPV infection process appears to be based on the integration of HPV DNA into the human genome. This induces a deregulated transcription of the viral oncogenes E6 and E7, chromosomal instability, and genetic or epigenetic oncogenic alterations, leading to dysplasia and, potentially, to carcinogenesis [57]. The transmission of HPV is facilitated by sexual intercourse through mucosal contact or secretions. Based on this, several studies in recent years have investigated the potentially increased risk in partners of patients with a diagnosis of penile cancer. A systematic review in 2017 by Mirghani et al. showed that, following the first observation studies that showed a geographic correlation between rates of cervical and penile cancers, several subsequent studies supposed a causality regarding the incidence of cervical cancer in partners of patients diagnosed with penile cancer. This suggested that partners of patients with HPV-related cancers also had a higher risk of developing HPV-related cancers [22]. However, this study was subject to various limitations, as indicated by the authors, and more recent studies did not show any higher risk in this population, and thus did not confirm this hypothesis [23,58]. Another interesting study about the role of sexual activity in the development of penile cancer was conducted by Madsen et al., which showed a significant association between heterosexual oral-penis sex and the risk of squamous PC. More specifically, the association with the number of female partners performing oral sex (OR 3.65; 95% CI, 1.14–11.7; for ≥3 versus 0 oral sex partners, *p* = 0.04) was statistically significant in the multivariate analysis. This suggested that oral sex could be an underestimated factor in HPV transmission increasing PC risk [24]. HPV is not the only sexually transmitted infection that can play a role in PC development; thus, some authors also considered HIV infection as a potential risk factor for PC or other HPV-related cancers. Although a clear mechanism has not yet been elucidated, HIV-related immunosuppression probably plays an important role [46]. A 2016 estimate showed that nearly 40 million people globally are living with HIV [59]. Its prevalence is concentrated mostly in the African territories, whereas, in low prevalence territories such as Europe, the virus is transmitted mainly in certain contexts, such as drug users, men who have sex with men (MSM), sex workers, and transgender people [59]. In addition, it appears to be an aggravating factor and an accelerator of disease progression [25,60]. Concerning predisposing factors of disease progression, as in other cancers, chronic inflammation can play an important role in creating a suitable microenvironment for PC development/progression. In this context, studies have recorded a significant increase in the risk of PC among males with phimosis [21]. Thus, during sexual arousal, the preputial epithelium stretches, thus diminishing its already thin layer of keratin. During intercourse, this vulnerable structure is directly exposed to the potentially infected partner’s secretions [60]. The preputial cavity offers a hospitable environment for an infectious inoculum; moreover, it can be the retention site of the epidermal cells, urine, and smegma, leading to chronic inflammation and possible bacterial superinfection [61]. The available studies regarding the effect of circumcision on the risk of contracting HPV are conflicting [62,63,64]. However, some papers reported a lower incidence of PC in Jewish men [65] and a recent meta-analysis described a strong protective effect of childhood/adolescent circumcision on invasive PC (OR 0.33; 95% CI 0.13–0.83) [66]. These data point to circumcision as an effective strategy for penile tumor prevention. No study correlating penile trauma or sex-related microtrauma to the onset of PC is available in the literature.

### 3.3. Testicular Cancer (TC)

TC is the most common solid tumor in males between 20 and 40 years, representing 1% to 1.5% of cancer in males, and 5% of urologic tumors in general [67]. Among TCs, testicular germ cell tumors (GCTs) represent 95% of malignant tumors, and are further clinically and histologically subclassified into seminomas and non-seminomas. Non-seminoma tumors are the more clinically aggressive subtype but, with current treatment, 5 year survival rates exceed 70% [68]. The pathogenesis of this cancer is not fully clarified, but the rapid increase in incidence suggests that critical changes in environmental factors may contribute to the development of cancer [69]. Several risk factors have been proposed for TC, including cryptorchidism, contralateral GCT, familial association, infertility, testicular atrophy, trauma, surgery, socioeconomic status, environmental factors, occupational exposure to noxious conditions, and Klinefelter syndrome [67]. Several trials involving controlled studies, cancer registry reports, clinical case series, and case reports have explored these factors. The data are relatively heterogeneous and the significance of some parameters remains uncertain. Undescended testis (cryptorchidism), contralateral testicular GCT, and familial testicular GCT are the only factors with an established evidence. Infertility, twinship, and testicular atrophy are also associated with GCT risk, but the level of evidence is clearly lower [70]. More recently, the literature has focused on the association between inflammation and cancer, suggesting that it may play an important role in the occurrence and progression of TC [5]. The relationship between inflammation and tumors can be partly attributed to infections, with up to 20% of all cases of cancer globally associated with microbial infections [5]. In particular, chronic inflammation has been founded to mediate several diseases, such as as cardiovascular diseases, cancer, diabetes, arthritis, Alzheimer’s disease, pulmonary diseases, and autoimmune diseases [71]. In tumorigenesis, chronic inflammation is involved in different steps, including cellular transformation, promotion, survival, proliferation, invasion, angiogenesis, and metastasis [72]. Inflammation cells, such as macrophages and other leukocytes, produce several mutagenic agents, such as reactive oxygen and nitrogen species, that cause mutation when they interact with DNA. Furthermore, DNA damage is increased by tumor necrosis factor-alpha (TNF-α) and macrophage migration inhibitory factors. The migration inhibitory factor is involved in the p53 pathway, which causes an accumulation of oncogenic mutations and alteration of the Rb-E2F pathway [73]. In the 1980s, Newell et al. and Algood et al. theorized the possible causal relationship between viral infections and TGCT [26,74]. Most of the viruses involved in STDs have an age-related prevalence that coincides with that observed in TGCTs [27]. However, the literature does not show a clear and direct link between SA and the development of TC. In recent years, several viruses potentially associated with the development of TGCTs (e.g., human papillomavirus (HPV), Epstein–Barr virus (EBV), cytomegalovirus (CMV), Parvovirus B-19, and human immunodeficiency virus (HIV)) have been studied, but with contrasting results [27]. The immune system plays a key role in the prevention of the occurrence of clinical malignancy; therefore, HIV may create a favorable environment for the development of tumors, including those of TC [28]. In Western countries, the majority of HIV-infected people are men, thus explaining why the incidence of testicular cancer is also increased in men with HIV [75]. In a 2017 review, Hentrich showed that seminoma and extragonadal germ cell cancer are more frequent in HIV-infected patients, whereas the risk for nonseminoma was marginally increased [76]. In the same way, HPV is closely linked with infertility in males and the role of this virus in testicular carcinogenesis cannot be excluded [66]. Recently, Kao et al. focused on epididymoorchitis, an inflammatory disease that affects the epididymis and the testis, caused by a wide range of microbes, including sexually transmitted and urinary germs. The authors found that patients with TC had a higher prevalence of prior epididymoorchitis than subjects without TC (11.0% vs. 0.3%, *p* < 0.001) and prior epididymoorchitis was significantly associated with TC even after adjustment for other variables (OR 47.17, 95% CI 23.83–93.40) [73]. However, a meta-analysis conducted by Traber et al. reported that mumps orchitis or orchitis infection were not associated with TGCT (pooled OR: 1.80, 95% CI 0.74–4.42) [30]. Testicular inflammation can also be the consequence of a traumatic event; more specifically, traumatic injuries to the testicle may result from either blunt or penetrating trauma. Sexual intercourse, in addition to sport activities, may represent a cause of this trauma. Testicular trauma can lead to various consequences, such as hydrocele, hematocele, hematoma, testicular rupture, alterations in perfusion, and dislocation of the testis [77]. No relationship between testicular trauma and cancer has been described in the literature; however, instrumental examinations indicated in these situations may enable the identification of pre-existing lesions [78].

## 4. Prevention

Although the relationship between sexual activity and male cancer is not yet fully understood, and strong evidence has not yet been provided, there are some behaviors that could be adopted to reduce risk. First, prevention of STIs can play a crucial role in decreasing cancer incidence [79]. Several measures can be adopted to prevent or reduce STI transmission:vaccination coverage; reduction in the efficacy of transmission (condoms, microbicides, etc.); treatment of the infection to reduce the duration of infectivity; changes in sexual behaviors; avoidance of oral, anal, and vaginal sex; having a mutually monogamous relationship with someone who is not infected; decreasing the number of sexual partners; and avoiding or limiting the use of alcohol and drugs before and during sex [79] (Table 2).

The use of male condoms is the most effective method of reducing STI transmission. Condoms provide an impervious barrier to pathogens in genital fluids. Their correct and consistent use is highly effective in reducing the risk of transmission of different STIs, including HIV/AIDS. Nonetheless, condoms may fail due to either method failure or user failure. Examples of the former include rupture or breakage, slippage, incomplete coverage of infected areas, and rare manufacturing defects. Types of user failure include inconsistent use, genital contact prior to use, incorrect application and positioning, use of oil-based lubricants, and incorrect withdrawal. Several studies have shown that the consistent and correct use of condoms reduces the risk of acquiring HIV infection by 87%. In addition, the importance of the correct and consistent use of condoms has been confirmed by data that show about 80% of new HIV transmissions result from individuals who do not know they are infected [80]. The most important health problems relate to adolescents that have a misconception of the correct and consistent use of condoms. One of the most important topics in public health is the introduction of an appropriate and accurate educational course. Vaccines against STIs, and particularly those against HPV, are another important topic in public health. At present, guidelines indicate that there are three vaccines approved in Europe for females and males [81], namely: Gardasil (Merck & Co., Kenilworth, NJ, USA), a quadrivalent HPV vaccine and the first commercially available HPV vaccine licensed by the United States Food and Drug Administration (FDA), in 2006; Cervivax, (GSK, Brentford, UK) a bivalent HPV vaccine, approved by the European Medicines Agency (EMA) in 2007 and by the FDA in 2009; and the World Health Organization’s human papillomavirus vaccines [82], which protects against the most common oncogenic genotypes of HPV (types 16 and 18), responsible for around 70% of cervical cancers [83]. Gardasil was developed against HPV 16, 18, and 11. In particular, HPV 6 and HPV 11 cause around 90% of genital warts. In 2014, a nine-valent vaccine, Gardasil 9 (Merck & Co., Kenilworth, NJ, USA), was licensed by the FDA, and offers protection against HPV 6, 11, 16, 18, 31, 33, 45, 53, and 58 [84]. According to the World Health Organization, HPV vaccination is recommended at the age of 9–14 in girls and, in some countries, boys aged 11–13 years [85,86]. Vaccination at a young age is preferred because several studies have shown that vaccines are less effective after the onset of sexual activity and exposure to HPV. Thus, it is strongly recommended that immunization takes place before initiation of sexual activity and subsequent exposure to HPV [87]. Although there is a lower risk of HPV-related cancers in heterosexual men, it is very important to vaccinate men because this offers protection for both individuals, thus leading to a faster achievement of “herd immunity” by reducing the size of the overall HPV reservoir [87]. Several studies have shown the positive effects of vaccines in men. The quadrivalent vaccine in HPV-naïve men has been shown to reduce the risk of a persistent anal HPV 16/18 infection by 96% [88]. Hillman et al. showed that the immunogenicity of the quadrivalent human papillomavirus vaccine is highly effective in men aged from 16 to 26 years, with seroconversion within 7 months and antibody detection even at 36 months [89]. Giuliano et al. showed similar outcomes regarding the effect of the quadrivalent HPV vaccine against infection in male patients [90]. In a recent systematic review of 5196 articles and seven studies (four randomized controlled trials (RCTs) and three non-randomized studies) involving 5294 participants, on the effectiveness and safety of vaccination against the human papillomavirus in male patients, Harder et al. showed that vaccine effectiveness was low in individuals who were previously infected with the corresponding HPV type, but high when compared with groups of HPV-negative males [91]. Thus, early vaccination of boys is appropriate, with the goal of vaccine-induced protection before the onset of sexual activity. Another important topic regarding the prevention of HPV infection is circumcision. Although data are still unclear, many studies have shown a positive effect of circumcision to prevent HPV infection. In a study involving 4033 healthy men, Albero et al. showed that circumcision is not associated with the incidence and clearance of genital HPV detection, with the exception of certain HPV types. The clearance incidence was significantly lower among circumcised versus uncircumcised men for HPV types 58 (*p* = 0.01), 68 (*p* < 0.001), 42 (*p* = 0.01), 61 (*p* < 0.001), 71 (*p* < 0.001), 81 (*p* = 0.04), and IS39 (*p* = 0.01), and higher for HPV types 39 (*p* = 0.01) and 51 (*p* = 0.02). Despite the lack of an overall association between the risk of HPV clearance and circumcision (for any HPV, aHR 0.95, 95% CI 0.88–1.02) [92]. Tobian et al. showed that the prevalence of high-risk HPV genotypes was 18.0% in the circumcised group and 27.9% in the uncircumcised group (adjusted risk ratio, 0.65; 95% CI, 0.46 to 0.90; *p* = 0.009) [93]. In another randomized controlled study involving 1264 men aged 18–24 years, Auvert et al. showed that the prevalence of urethral HR-HPV infection after circumcision was reduced (95% CI 1.5–3.3) and 2.2 (95% CI 1.5–3.2) [94]. Circumcision has various health benefits, not only related to HPV prevention, namely, making it easier to wash the penis, decreasing the risk of urinary tract infections in males, and decreasing the risk of penile cancer [95]. Circumcision is highly important in phimosis prevention and treatment. Phimosis or lichen sclerosus can cause chronic preputial inflammation, which, in several cases, is associated with penile cancer. Circumcision lowers the risk of penile cancer (hazard ratio: 0.33) [96].

## 5. Strengths and Limitations

To the best of our knowledge, this is the first systematic review describing the impact of SA on male genital cancers. This paper should be interpreted in the context of several limitations. The main limitation is the lack of data synthesis. Other weaknesses are the relative paucity of relevant studies (particularly for TC), and the studies’ heterogeneity, contradictory results, and overall moderate-to-low quality. In addition, several mechanisms by which SA may be implicated in the development of male genital tumors are only hypothesized or inferred indirectly. Finally, the lack of detail in the descriptions of the characteristics of SA (e.g., type, duration, frequency, and partner) in the original studies makes it difficult to interpret the data.

## 6. Conclusions

The association between SA and male genital tumors may be based on several sex-related risk factors. SA is closely associated with the development of PC through the high-risk HPV transmission; in this context, phimosis appears to be a favoring factor. Sexual behaviors appear to play a significant role in PCa pathogenesis, probably through inflammatory mechanisms; however, protective sexual habits have also been described. A direct correlation between SA and TC has not yet been proven, although infections remain the most studied sex-related factor. The limited evidence available in the literature prevents clear conclusions from being drawn. Further primary studies with appropriate designs are needed to elucidate the association between SA and male genital tumors.

## Figures and Tables

**Figure 1 ijerph-18-08500-f001:**
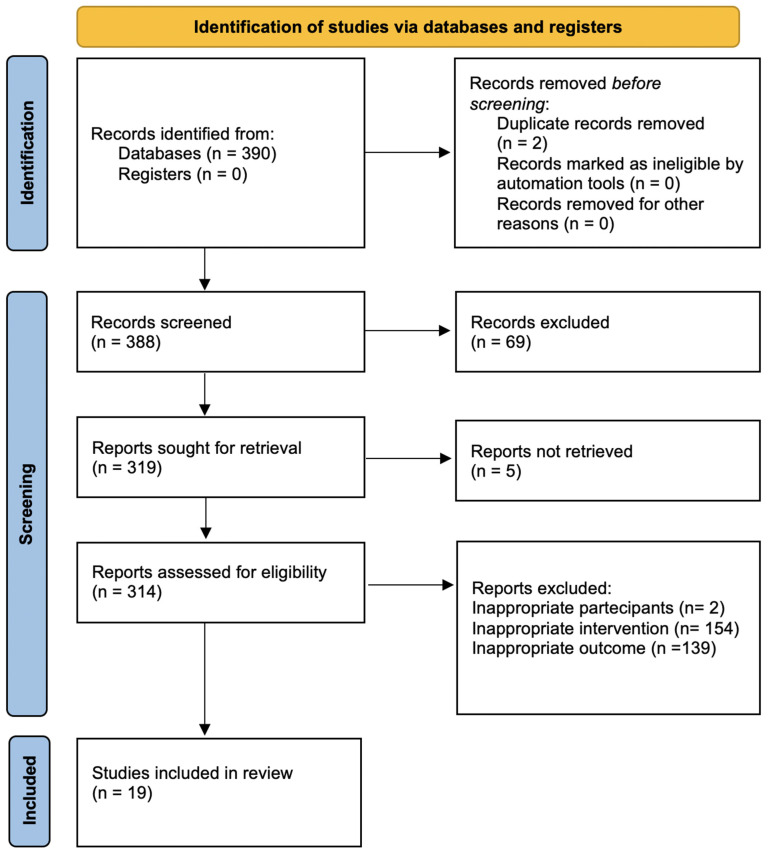
PRISMA flow diagram for selection of studies.

**Figure 2 ijerph-18-08500-f002:**
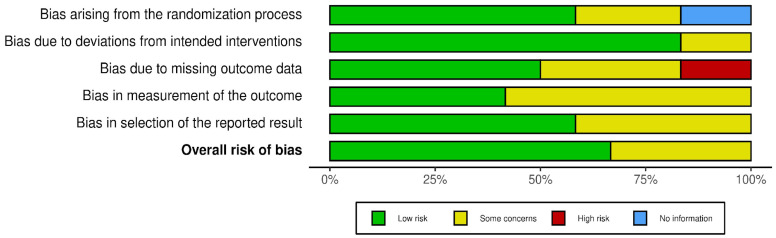
Risk of bias.

**Figure 3 ijerph-18-08500-f003:**
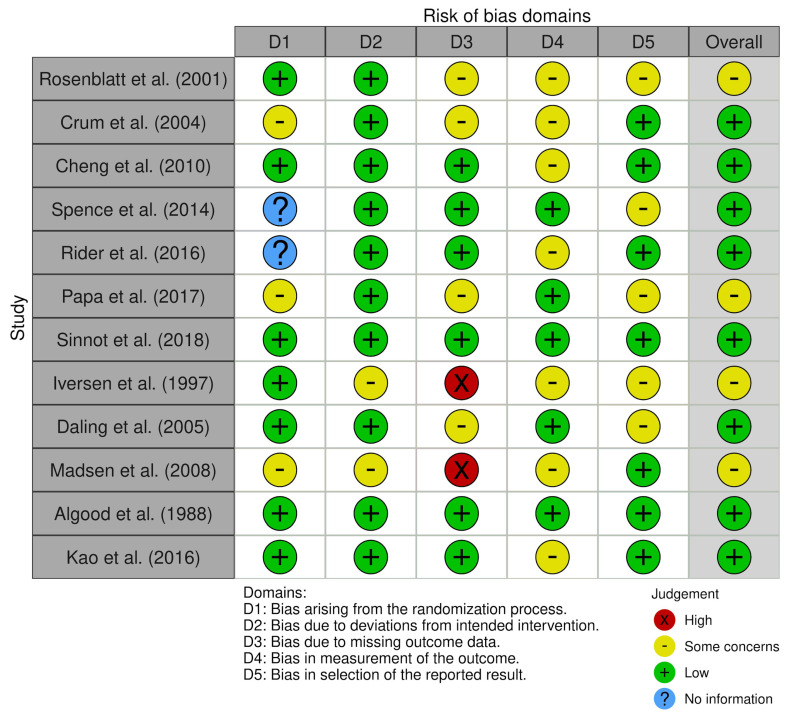
Risk of bias domains.

**Figure 4 ijerph-18-08500-f004:**
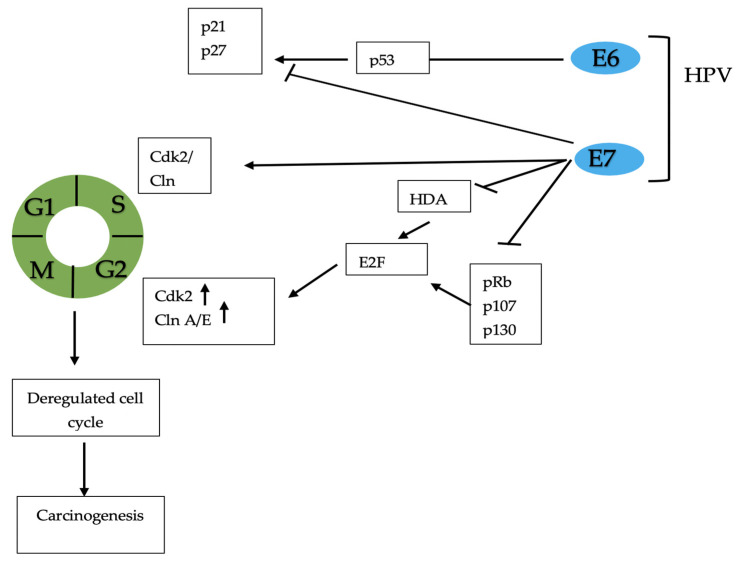
Molecular mechanisms of HPV-induced carcinogenesis. E6 and E7 bind and inactivate the p53 pathway, and pRb, p21, p27, p107, and p130 tumor suppressor gene products. This leads to cell cycle disregulation and carcinogenesis. HPV: human papillomavirus.

**Table 1 ijerph-18-08500-t001:** Main characteristics of the selected studies.

First Author and Publication Year	Study Design	Sample Size	Main Findings	Study Quality
Prostate Cancer				
Rosenblatt 2001 [16]	Case-control	1456	Association between sexual factor and PCa	5 ^#^
Spence 2014 [12]	Case-control	3208	Reduction of risk of PCa in men with >20 sexual partners No correlation between Risk of PCa and age at first sexual intercourse No association between STDs and PCa	8 ^#^
Papa 2017 [14]	Case-control	2141	Inverse associations with EF and PCa	6 ^#^
Crum 2004 [20]	Prospective cohort study	269	Positive correlation between incidence of PCa and duration of HIV infection	6 ^#^
Cheng 2010 [18]	Prospective cohort study	68,675	Positive correlation between prostatitis, STDs and Pca	8 ^#^
Rider 2016 [13]	Prospective cohort study	31,925	Inverse association between EF and PCa	7 ^#^
Sinnott 2018 [15]	Prospective cohort study	157	Inverse association between EF and PCa	6 ^#^
Jian 2018 [17]	Systematic Review	55,490	Positive association between number of female sexual partners, age at first intercourse, EF, and the risk of PCa	Intermediate *
Sun 2021 [19]	Systematic Review	2780	Reduced risk of PCa among people with HIV/AIDS	High *
Penile Cancer				
Iversen 1997 [23]	Case-control	867	Not significant increased of incidence of HPV related cancers in partner of patients with HPV related cancer	5 ^#^
Daling 2005 [21]	Case-control	808	Positive association between HPV and penile cancer	7 ^#^
Madsen 2008 [24]	Case-control	293	Positive correlation between oral sex as a risk factor in PC	5 ^#^
Mirghani 2017 [22]	Systematic Review	1356	Higher incidence of HPV related cancers in partner of patients with HPV related cancer	Low *
Lekoane 2020 [25]	Systematic Review	16,351	Positive correlation between HIV andHPV-related cancers	Intermediate *
Testicular Cancer				
Algood 1988 [26]	Prospective Cohort study	86	Positive correlation between Viral infections (Epstein-Barr virus, cytomegalovirus, and hepatitis A and B viruses) and TGCTs	6 ^#^
Kao 2016 [29]	Case-control	4092	Positive association between TGCTs and Epididymoorchitis	8 ^#^
Garolla 2019 [27]	Meta-analysis	285,878	Positive correlation between viral infections and risk of developing TGCTs	High *
Trabert 1969 [30]	Meta-analysis	1696	No association between common infections and TGCTs	Intermediate *
Grulich 2017 [28]	Meta-analysis	476,149	Higher incidence of cancers in people with HIV/AIDS compared with immunosuppressed transplant recipients	High *

EF: ejaculatory frequency; HIV/AIDS: Human immunodeficiency Virus/Acquired Immune Deficiency Syndrome; HPV: human papillomavirus; PCa: prostate cancer; STDs: sexually transmitted disease; PC: penile cancer; TGCTs: testicular germ cell tumors. * A Measurement Tool to Assess Systematic Review-2 (AMSTAR-2), ^#^ Newcastle–Ottawa Scale (NOS).

**Table 2 ijerph-18-08500-t002:** Role of SA in development of male genital cancers.

	Protective Factors	Risk Factors
Penile cancer	Circumcision *	Transmission of HPV 16 and HPV 18; transmission of HIV; phimosis
Prostate cancer	High ejaculatory frequency	High number of sexual partners; homosexuality; STDs (e.g., gonorrhea)
Testicular cancer	Trauma **	Viral infection and epididymoorchitis ***

SA: sexual activity; HPV: human papillomavirus; STD: sexually transmitted disease; NA: not available. * Conflicting data, ** Allows incidental diagnosis, *** Very low evidence.

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
