# Peer review of "Impact of Sexual Activity on the Risk of Male Genital Tumors: A Systematic Review of the Literature"

_ijerph, 2021, doi:10.3390/ijerph18168500_

Round 1
Reviewer 1 Report
It was my pleasure to review this manuscript, which is an exploratory and non-systematic review, which attempts to clarify the impact of sexual activity on the risk of male genital tumors.
The subject seems very interesting to me since male genital tumors are a great concern for our society.
Below I am going to make a series of comments to improve the quality of the manuscript:
The title should state the type of review in question. In this case we are facing an exploratory review.
The material and method section is very brief. Although it is not a systematic review, I believe that a minimum of methodology should be followed.
A table with a well detailed search strategy should be added.
The article selection process must be indicated following the PRYSMA method.
In the results section, the number of studies selected will be indicated and a table with the main characteristics of these studies will be added later.
The methodological quality of the selected articles should be evaluated by applying the GRADE method or another with similar characteristics and the results should be shown in a table.
The risk of bias present in the selected studies should also be studied and visually reflected by a figure.
In the limitations, the authors acknowledge that the design is not systematic and that data synthesis is lacking.
I believe that currently this exploratory review does not have sufficient methodological quality to be published in this journal.
I regret that I cannot give a more favorable assessment at this time.
Thanks
Kind regards
Reviewer 2 Report
Thank you for submitting this manuscript for consideration. You have chosen to focus on a relevant and contemporary topic in relation to male genital cancers. Whilst you do acknowledge the limitations of this review, I would suggest that further work is required prior to consideration for publication.
The choice of topic is of interest within healthcare and public health.
Search strategy - this requires some expansion and justification, particularly when suggesting a narrative review approach. Why did you choose those databases for the search? How was the quality of the papers considered for their inclusion within the review? Where there other objectives in order to meet the aim of the review?
Results are presented under headings of the cancers rather than under a considered theme. This makes the paper read more like an opinion or information piece than a narrative review and appraisal of the current knowledge.
Conclusions are satisfactory.
Strengthening the review in this manner would encourage more interest in it. It would also strengthen any recommendations made for future practice or research.
Round 2
Reviewer 1 Report
It was my pleasure to review this second improved version of the manuscript.
The authors have endeavored to respond to and apply the reviewers' recommendations, notably improving the quality of the manuscript.
The manuscript seems correct to me for its publication in this Journal and I have no objection to it.
I think a good job was done.
Thank you
Kind regards
Reviewer 2 Report
Thank you for submitting this revised version and your comments. These have been appropriately addressed and I do not have any further comments to offer for the revised manuscript. I have enjoyed reading this work as the topic is interesting and relevant.